# Modeling Formation and Operation of Collaborative Green Innovation between Manufacturer and Supplier: A Game Theory Approach

**Qian Li** [1,*] , **Yuanfei Kang** [2] , **Lingling Tan** [3] and **Bo Chen** [1]

1   School of Economics and Management, Chang'an University, Xi'an 710064, China; chb@chd.edu.cn
2   Massey Business School, Massey University, Auckland 0745, New Zealand; Y.Kang@massey.ac.nz
3   School of Modern Post, Xi'an University of Posts and Telecommunications, Xi'an 710061, China; tanlingling1986@chd.edu.cn
*   Correspondence: laplace0911@163.com

**Abstract:** Prior research has mainly emphasized the strategic importance of a collaborative green innovation (CGI) between the manufacturer and supplier in a supply chain, leading to an overlook at the decision-making mechanism and determinants of CGI. Guided by the transaction cost economics and social exchange theory, our study constructs a mathematical game model to incorporate the key dimensions of an effective inter-firm collaboration for green innovation. Applying the Nash game bargaining principles, our evolutionary game model analysis provides an analytic system to understand the mechanisms of forming and operating a collaboration partnership between the manufacturer and supplier for green innovation. Based on various scenarios from the numerical simulation parameters for the involved influencing factors, our simulation has produced the Nash equilibrium solutions and identified the major determining factors for successfully forming and operating CGI. They are the trust level between the manufacturer and supplier as the CGI partners, value/profit sharing ratio between the partners, knowledge complementarity of the partners, and product type for the green innovation.

**Keywords:** collaborative green innovation; knowledge complementarity; revenue distribution; trust level; Nash bargaining game

## 1. Introduction

Research in green supply chain management has identified suppliers, especially the strategic suppliers, as the main source for resource consumption and environmental pollution for manufacturing products throughout the whole process of product design and development, as well as distribution and transportation [1,2]. As a result, a manufacturer's green innovation capability lies in its ability to involve effective participation of suppliers and coordinate internal as well as external resources for green innovation [3]. Prior research suggests that collaborating with suppliers for green innovations plays a critical role in achieving commercial performance for the involved firms and environmental performance for the consumers and the society [4,5], by means of sharing technology and green knowledge, sharing the resources and facilities, and setting standards for green products and processes. Thus, in responding to the trend of green innovation in the manufacturing industry, manufacturing firms have been increasingly engaging in the strategy of collaborating with their suppliers for green innovation [4,6–8].

Prior studies have examined various issues with regard to facilitating and improving collaborative green innovation (CGI) between supply chain partners, such as optimizing supply chain power

structures [9], providing government subsidies to customers [10], and providing government incentives to the firms [11]. While the supply chain relationship evolves from purely market-based transactional behavior to the long-term collaborative partnership behavior, research attention regarding supplier chain relationship has gradually shifted from minimizing transaction costs and/or maximizing transaction profits to building long term collaborative partnership based on social exchange theory [4,12–14]. The expected profit creation from green innovation has been considered as the crucial motivating factor to CGI [5,15–17]. Meanwhile, the game theory approach has been widely applied as a modelling analytic framework to the various areas regarding profit sharing in supply chain, such as decision-making patterns in relation to cooperative versus non-cooperative behavior between the manufacturer and supplier in responding to regulatory policy on carbon dioxide reduction [18], dynamic game modeling analysis regarding effect of inter-firm completion on technology sharing strategy [19], knowledge sharing mechanism for collaborative innovation by supply chain members [20], and Stackelberg game model analysis regarding effect of knowledge spillover on inter-firm resource sharing.

On the other hand, some important issues regarding collaboration between the manufacturer and supplier in green innovation have still not been addressed. Firstly, while CGI needs to be built on the supply chain partnership [21], it is still not clear about what are the key motivating factors that facilitate formation of supply chain green collaboration [22,23]. More specifically, formation and operation of such collaboration relies on incorporating various factors, including mutual trust, knowledge sharing, mutual learning, effective inter- and intra-firm coordination, and profit distribution mechanisms. Thus, when making decision to CGI, both parties need to incorporate these factors in their consideration. Secondly, while there is consensus that green innovation would generate commercial and social benefits for the relevant stakeholders, the commercial value generated from CGI may take a long term to realize for the involved firm, and is likely to be retained by the manufacturer [4,6]. Thus, it is necessary to design a fair distribution policy about collaborative green innovation. Thirdly, various game models, such as evolutional game model, prisoner's dilemma game model, and Stackelberg game model, have been applied to analyze inter-firm collaborative behavior in making binary choice, such as cooperative versus non-cooperative in supply chain, knowledge sharing versus not-sharing [18,24,25]. However, existing research has neither investigated issues relating to collaboration level in CGI formation, nor incorporated quantified influencing factors, such as mutual trust, mutual learning, and profit distribution ratio in the game models.

To address the research gaps identified above, we develop a game model for decision-making on forming a collaborative green innovation and for analyzing its evolution under various scenarios. Following existing game model conceptualization for collaborative product innovation [21], our model incorporates four key dimensions of effective inter-firm collaboration of green innovation, including trust level, collaboration level for green innovation, cost sharing ratio, and profit distribution ratio. Appling the Nash game bargaining principles, our simulation analyses the Nash equilibrium solutions based on the various scenarios from the numerical parameters for the relevant influencing factors, and thus provides a decision support system to understand the mechanisms of decision-making to form a collaboration partnership between the manufacturer and supplier for green innovation. In summary, our study contributes to the existing literature by providing a decision support system for the manufacturer and supplier on how to develop their decision-making strategy when negotiating a CGI partnership and collaborating with their respective partner firms by providing the equilibrium solutions.

The paper is organized as follows: Section 2 provides theoretical foundation and discusses factors selection. Section 3 introduces the game theory applications in the CGI and describes our mathematical model for CGI. Section 4 presents our numerical simulation analysis for several scenarios of the collaboration to numerically illustrate the details of the game model and to make inferences. The last section provides concluding remarks and practical implications.

## 2. Theoretical Foundation and Factor Selection

### 2.1. Green Innovation and CGI

Green innovation is an innovation activity that aims to protect the natural environment by reducing or avoiding environmental pollution, in order to enable the firm meet new market demands, and to create green value [26]. Green innovation practice is an important way for the firm to increase its creativity and improve the efficiency of resource utilization. By engaging in green innovations, firms can thus gain the advantages of brand/firm image, new product development, enhanced competitive position [27,28], and improved financial performance [29]. Green innovations include three dimensions of green product innovation, green process innovation, and green management innovation. Green product innovation refers to the improvement of the existing product design process to reduce the negative environmental impact of the product in its lifetime; green process innovation refers to the adjustment of the manufacturing process to reduce the material acquisition, production, and transportation process to reduce the negative impact on the environment [5,29]; green management innovation emphasizes the success of green management [30] through the support and commitment of senior management.

Collaborative innovation is the activity of multiple member companies to create sustainable competitive advantage by sharing common goals, trust, respect, resources, green knowledge, risks, and rewards, thereby achieving greater benefits than acting alone [31]. Suppliers involved in CGI have several forms from providing small design suggestions to develop a specific component, or even designing a complete component or project [26,32]. The collaboration level depends on different levels of partnership, motivation factors, and contribution to the output of manufacturer [33,34].

### 2.2. Motivation Factors for Suppliers to Participate in Green Innovation Cooperation

Success of CGI in the supply chain results from the accumulation of internal and external knowledge of various involved firms within the supply chain [35]. The factors enabling a firm's participation in CGI are identified as R&D motivation, learning motivation, strategic motivation, and competition motivation. Although, suppliers' participation in manufacturers' green innovation is a knowledge cooperation activity with high failure rate, high knowledge spillover risk, and high opportunistic risk, the participation motivation can be explained as the following reasons. One is supplier's expectation of maintaining long-term strategic partnerships to prevent competitors from intervening, and in turn, strategic partnerships with manufacturing enterprises can promote opportunities for green innovation or environmental cooperation [8]. The other is knowledge searching and cooperative learning. As manufacturers tend to have strong technical capabilities, innovative capabilities, and extensive market information, suppliers would be learn from the advanced technologies, acquire knowledge, and improve their innovation capabilities through participating in CGI.

### 2.3. Determinants for Suppliers Involvement into CGI

Different from research on cooperative behavior, the process of CGI focuses more on the dynamic interactions between the two partners. Thus, the success of CGI with supplier should be based on mutual partnership, mutual learning, collaboration level, green innovation value, and profit distribution strategy [23,36]. They are discussed as follows.

(1)　Partnership of mutual trust.

Trust is the cornerstone of business partnerships, and it nurtures the intention of knowledge acquisition and sharing outside organizational boundaries [37]. Empirical evidence suggests that trust is a necessary condition for interorganizational knowledge learning in green supply chain [38], as it plays an essential role for forming and sustaining an effective collaboration in green innovation [39,40]. Empirical evidence confirms that relational capital in the supply chain network can significantly strengthen management of CGI in the supply chain and improve performance of the involved

firms [7]. Suppliers' participation of the CGI means forming a R&D team and sharing their appropriate green knowledge with the manufacturing firm, but knowledge sharing may generate risk of knowledge leaking. When considering the risk that the manufacturer could plagiarize its appropriate knowledge/technology or unilaterally modify the conditions for green collaboration, a rational supplier could be less willing to participate collaboration. Moreover, parties involved in the collaborative partnership have to commit more resources and cover the extra costs to closely monitor the collaboration process in order to prevent knowledge leaking [39]. Trust between collaborative partners and its related social control mechanism would increase the probability of building a long-term partnership by influencing integration of supply chain partners [40,41]. Thus, trust is identified as an important factor influencing decision-making of knowledge sharing, as well as the supplier's participation in CGI [4,8]. Based on the trust–commitment theory, a successful partnership requires commitment among the partners, and trust is a critical element to sustain such commitment. Thus, the trust level can be measured by transaction cost variables, including asset specificity, behavioral uncertainty, and social exchange variables including perceived satisfaction, reputation, and conflict [42].

(2)　Knowledge complementarity and mutual learning.

Learning about complementary knowledge plays an important role in successful collaboration innovation [43]. A firm's learning about the knowledge that is complementary to its existing knowledge enhances its knowledge structure and is thus able to facilitate more effective innovation [44]. Mutual learning in a partnership promotes an integration of the separate knowledge possessed by the involved individual partners, and thus enables application of the relevant knowledge in the innovation practice [45]. Just because knowledge is related with each other, and thus learning becomes a less difficult task. As a result, mutual learning built up on the complementary knowledge can in turn promote collaboration and strengthen the partnership. Empirical research has also demonstrated that knowledge complementarity is one of the driving forces to inter-firm collaboration, as it enables collaborative partners' assimilation and absorption of knowledge accessed from the partner. In the context of CGI, mutual learning between the supplier and manufacturing enterprise facilitates an effective use of knowledge and technological resources within the partnership, leading to continuous creation of new green knowledge and development of new green products. In a word, the level of mutual learning is dependent on knowledge complementarity, total knowledge stock, and absorptive capacity of the collaborative partners [41], and knowledge complementarity serves as the bridge between knowledge sharing and knowledge creation [46].

(3)　Level of collaboration.

CGI requires the involved supplier and manufacturing enterprise to establish mutually agreed goals for green innovation. Thus, level of collaboration is another key factor for CGI [47], as various work teams probably at different sites are engaged for the R&D activities, so the joint innovation efforts need to be coordinated and synchronized in order to achieve the mutual goals. Traditionally, different levels of collaboration would result in corresponding levels of coordination [48]. Green innovation cooperation requires suppliers and manufacturers to have the same environmental goals and responsibilities, they need work together in joint planning and joint decision-making to solve green problems [31,48]. The collaboration also requires people from different organizations in different locations to have synchronous participation in green innovation activities. Thus, cooperation between green innovation teams is the key to the success of CGI. Empirical research has shown that firm's capabilities in information technology (IT) have a significant and positive effect on knowledge sharing and cooperation performance [49]. Thus, development and/or improvement of firm IT capabilities, including capabilities in technology infrastructure, information and communication technology, integration technologies, and green product development platforms, play a critical role in promoting mutual learning and facilitating effective inter-firm collaboration. However, development of IT capabilities requires significant resource input and a high level of cooperation between the involved partners. That is, high level of collaboration incurs higher cooperation costs.

(4)　　Green added value and innovation efficacy.

Output from green innovation could be measured from several dimensions [26], including: (1) Reduction of waste and environmental impact, such as reduced pollutant emissions, use of renewable energy, using environment-friendly materials, and avoiding use of toxic substances; (2) improvement of the market performance (such as entering into new markets) and establishment of competitive advantage; (3) enhanced corporate image performance (such as increased reputation and improved regulatory compliance); (4) improvement of product green performance by using green materials or clean energy; (5) improved firm financial performance, such as increased revenue or market share, given that customers are willing to pay a higher price for green products. Taking green new product development as an example, successful green innovations would enable manufacturing enterprise to increase product greenness and generate green added value [50], thereby achieving high returns and high profit [51].

(5)　　Green revenue distribution strategy.

Green revenue is be one of the major goals pursued by both the supplier and manufacturer in CGI activities, so effective innovation profit distribution strategy (profit distribution ratio) is an important factor affecting the formation for CGI, as well as the overall innovation value. Researchers have found that it is also of great significance for improving the coordination between supply chain members and promoting the stability of cooperation [7].

## 3. Modelling Green Innovation Collaboration Formation

### 3.1. Background of the Model and Assumptions

In this section, we present a mathematical model based on the game theory approach to support firms in decision-making regarding CGI. Background of the model is as follows: Firms collaborate on their efforts of green innovation to maximize their revenue. As the leading firm in the supply chain, the focal manufacturer F would expect to collaborate with its main supplier S on development of the green technology, so as to add the green value to their products and improve the economical and environment performance. We consider revenue sharing as the basis for decision-making regarding the collaboration. As revenue is expected to generate from CGI, the two partners must decide on the collaboration level and investment sharing. For different levels of innovation collaboration, the decision-making process between the two partners is a continuous dynamic game, and the collaboration revenue can be optimized by adjusting the revenue distribution ratio. The decision-making process is identified as follows: Firstly, the manufacturer F determines the green development project and the best level of green innovation. Then, the supplier S proposes a cost sharing plan for innovation input, and the two partners jointly determine profit distribution strategy through Nash bargaining game. Finally, the two partners carry out the optimal level of input and carry out innovation scheme to achieve the added green value.

Model assumptions: Based on the existing studies on collaboration formation [21,51,52], our game model is built on the following assumptions.

(1)　The green innovation revenue generated from the collaboration is calculated by the added income from the collaborative innovation minus the innovation cost, and the added value is incurred from coordinated efforts and knowledge spill-overs.

(2)　The two partners obtain the green knowledge stock based on the initial cost input of green knowledge, and they determine the level of knowledge sharing according to the level of trust granted to each other.

(3)　CGI could reduce the overall cost of green innovation, but collaboration still incurs a cost for the involved partners. CGI incurs costs, which include the knowledge input cost, green innovation development cost, and collaboration cost. As two partners need to invest in green

development, we assume that collaboration can improve innovation efficiency and reduce overall green development costs compared with independent green innovation. Here, the collaborative cost refers to the capital investment cost for information technology capability enhancement and the additional integration cost for the division of the development [53], which needs to be shared by the two partners.

### 3.2. Description of the Model

A number of variables that affect the manufacturer-supplier CGI are incorporated into our model. Table 1 provides a description of the variables, notations, definitions, and functions, which are used in the study.

**Table 1.** Notations, variables, and functions.

| | Notations/ Variables | Description |
|---|---|---|
| **status symbols** | $i = F, S$ | *F* refers to the leading manufacturer, *and S* refers to the supplier. |
| | a | Initial value of the product (a > 0). |
| | b | The added green value generated by CGI, defined as the marginal benefit of each unit of sales of green innovative products. |
| | β | The complementarity of green knowledge between the two firms. |
| | $t_{FS}$ | The manufacturer's trust level towards the supplier. |
| | $t_{SF}$ | The supplier's trust level towards the manufacturer. |
| **status variables** | $z_F$ | The stock of the manufacturer's green innovation knowledge. |
| | $z_s$ | The stock of the supplier's green innovation knowledge. |
| | $q_F$ | The manufacturer's knowledge input of green innovation (efforts). |
| | $q_S$ | The supplier's knowledge input of green innovation (efforts). |
| **control variable** | θ | Green collaboration level. |
| | κ | Distribution ratio of collaboration costs. |
| **independent varible** | ω | Green profit distribution ratio for manufacturer F. |
| | $\gamma_F(\ ), \gamma_S(\ )$ | Knowledge absorptive capacity of manufacturer F and supplier S. |
| | $L_F(\ ), L_S(\ )$ | Learning level of Manufacturer F and supplier S. |
| | $\overline{z}(\ )$ | Total knowledge storage owned the two firms for green innovation. |
| **function notations** | $c_F(\ ), c_S(\ )$ | Green development costs of manufacturer F and supplier S. |
| | $h(\ )$ | Cooperative cost function for CGI. |
| | $\overline{v}(\ )$ | The efficiency of successful green innovation development |
| | $\pi_F(\ ), \pi_S(\ )$ | Green innovation revenue for manufacturer F and supplier S. |

(1) Definition of absorptive capacity function $\gamma_I$, mutual learning level function $L_I$ and knowledge stock function $\overline{z}$.

Knowledge absorptive capacity $\gamma_i$ is defined as the competence of a firm to acquire, absorb, assimilate, and utilize the knowledge shared by partners. Firms i can enhance their knowledge absorptive capacity through improving green knowledge input $q_i$, such as more recruitment of R&D staff, more skills training, and more investment in green technology. Thus, a first-order partial derivative $\frac{\partial \gamma_i}{\partial q_i} > 0$ can be established. Meanwhile, as the effect of knowledge input on the knowledge absorptive capacity tends to increase at a constant or decreasing rate, so a second-order partial derivative $\frac{\partial^2 \gamma_i}{\partial q_i^2} \leq 0$ could be deduced.

The higher knowledge complementarity β means the greater knowledge difference between the two firms. When knowledge complementarity is high, mutual learning may become more difficult, so the first-order partial derivative $\frac{\partial \gamma_i}{\partial \beta} < 0$ is established. However, when the level of knowledge complementarity is high, a firm's learning from its partner would be more effective in comparison to the case that the two firms possess similar knowledge. Thus, when knowledge complementarity β increases the marginal effect of knowledge input on absorptive capacity would increase. That is to say,

a second-order partial derivative $\frac{\partial^2 \gamma_i}{\partial q_i \partial \beta} \geq 0$ is established. Thus, the absorptive capacity function of the manufacturer and its supplier could be expressed as $\gamma_F(q_F, \beta)$, $\gamma_S(q_S, \beta)$.

The mutual learning level $L_i$ is defined as the level of knowledge learning, assimilating, and creating occurred during collaboration. $L_i$ is related to the level of green innovation collaboration $\theta$, the level of knowledge sharing, and absorptive capacity $\gamma_i$. In turn, the level of knowledge sharing depends on knowledge input in green innovation by the two firms and the trust level between them, because the trust level determines the degree of knowledge sharing between the two partners. Here, we define $t_{SF}$ as the supplier's trust level towards to the manufacturer, and $t_{FS}$ as the manufacturer 's trust level towards to the supplier. We confine $t_{SF}, t_{FS} \in [0, 1]$. Since innovation collaboration is able to improve the firm's learning ability, the mutual learning level can be defined as a convex function of the collaboration level $\theta$, which is a control variable. At a given collaboration level of $\theta$, the mutual learning level between the manufacturer and the supplier can be expressed as:

$$L_F = \theta \gamma_F(q_F, \beta)(q_S) t_{SF}$$

$$L_S = \theta \gamma_S(q_S, \beta)(q_F) t_{FS}$$

The total stock of green innovation knowledge possessed by the two firms is determined by their knowledge input to green innovation $q_i$ and the mutual learning level $L_i$ of the two firms. When the level of green innovation collaboration is $\theta$, the knowledge stock of the manufacturer and supplier can be expressed as functions (1) and (2), respectively, and the total stock of knowledge for green innovation can be expressed as function (3).

$$z_F = q_F + \theta \gamma_F(q_F, \beta)(q_S) t_{SF} \tag{1}$$

$$z_S = q_S + \theta \gamma_S(q_S, \beta)(q_F) t_{FS} \tag{2}$$

$$\overline{z} = q_F + q_F + \theta \left\{ \left[ \gamma_F(q_F, \beta)(q_S) t_{SF} \right] + \left[ \gamma_S(q_S, \beta)(q_F) t_{FS} \right] \right\} \tag{3}$$

(2)　Definition of green innovation development cost function $c_i$.

Through a specialization of innovation activities, innovation collaboration between the partners can lead to effective reduction of the total cost of innovation and improvement of the innovation efficiency. On the one hand, as the green knowledge input $q_i$ increases, the green knowledge stock $z_i$ of the two partners would increase correspondingly, leading to improvement of resource utilization and management efficiency in the innovation process, and then the costs for green innovation would be reduced. Therefore, $\frac{\partial c_i}{\partial z_i} < 0$ is established. On the other hand, as reduction of green innovation cost usually occurs at either a constant or a decreasing rate, $\frac{\partial^2 c_i}{\partial z_i^2} > 0$ is established. This is because CGI still incurs additional cost, although the total knowledge stock is increased by pooling the knowledge input from collaborating. Thus, the green innovation cost $c_i$ is a finite positive value, and $\frac{\partial^2 c_i}{\partial z_i^2} > 0$ is established.

(3)　Definition of cooperative cost function $h(\theta)$ and cost-sharing ratio $\kappa$.

The level of CGI determines the degree of information sharing, knowledge, and experience between collaboration partners. The higher collaboration level, the more information, knowledge, and experience that need to be shared in the collaboration process, and correspondingly, the cost of cooperation will also be higher. As a result, the cost of cooperation is usually defined as the quadratic function of the collaboration level $\theta$, it is expressed as $h(\theta) = I\theta^2$. Distribution by the manufacturer and the supplier for collaboration cost is expressed as $(1-\kappa)$ and $\kappa$, respectively, and both are included in our study as the control variables.

(4)　Definition of green innovation value b, innovation conversion efficiency $\overline{v}$, and profit distribution coefficient $\omega$.

Before carrying out CGI, the two partners would estimate the innovation efficiency $v$, by taking consideration of technical and environmental uncertainty. Due to the high failure rate of CGI, it is uncertain whether the green investment would generate the anticipative return. Following the study by Bhaskaran and Krishnan [51], we construct the innovation transformation probability $\bar{v}$ as the uniform distribution function of innovation efficiency $v$, expressed as $\frac{v+1}{2}$. The reason is that larger innovation efficiency $v$ creates higher conversion efficiency.

In order to simplify the model, we use green added value $b$ to evaluate green innovation output. In the environment of mutual trust and efficient collaboration, anticipative green added value can be defined as a function of green knowledge accumulation, and it is expected that there is a positive relationship between trust and green innovation efficiency. Thus, the green value added to the original products through green innovation is set as $b\bar{v}\bar{z}$, and the total value of green innovative products is set as $a + b\bar{v}\bar{z}$, where $a = 0$ represents green new product development, and $a > 0$ represents green improvement product development. $\omega$ is defined as the profit distribution ratio of CGI between the manufacturer and supplier, which is an independent variable.

### 3.3. Mathematical Model

Based on the above discussion, we define the green innovation revenue function for the manufacturer and supplier as follows:

$$\pi_F = \omega(a + b\bar{z}\bar{v}) - (1-k)h(\theta) - q_F - c_F(z_F) \tag{4}$$

$$\pi_S = (1-\omega)(a + b\bar{z}\bar{v}) - kh(\theta) - q_S - c_S(z_S) \tag{5}$$

As the estimation of the innovation conversion efficiency is $\bar{v}$, the anticipative value of green innovation revenue functions for the manufacturer and supplier are expressed as follows:

$$\pi_F = \omega\left(a + b\bar{z}\frac{v+1}{2}\right) - (1-k)h(\theta) - q_F - c_F(z_F) \tag{6}$$

$$\pi_S = (1-\omega)\left(a + b\bar{z}\frac{v+1}{2}\right) - kh(\theta) - q_S - c_S(z_S) \tag{7}$$

In order to further explore the relationship between various factors and the revenue function, we can simplify the function according to the function characteristics. The function of absorptive capacity for the manufacturer is interpreted as $\gamma_i(q_i, \beta) = q_i\beta + (1-\beta)$. When the collaboration level is $\theta$, the function of mutual learning level between the manufacturer and the supplier is : $L_F = \theta(q_F\beta + (1-\beta)) \cdot q_S \cdot t_{SF}$, $L_S = \theta(q_S\beta + (1-\beta)) \cdot q_F \cdot t_{FS}$. Thus, the total knowledge stock of green innovation possessed by the two firms is expressed as $\bar{z} = q_F + q_S + L_F + L_S$. Let $m_F = (q_F\beta + (1-\beta)) \cdot q_S \cdot t_{SF}$, $m_s = (q_S\beta + (1-\beta)) \cdot q_F \cdot t_{FS}$. The total stock of the green innovation knowledge of the two firms can be expressed as $\bar{z} = q_F + q_S + \theta(m_F + m_s)$.

The total development costs of CGI are expressed as $c = (c_S + c_F)$, which would reduce with the increase of the knowledge stock. The coefficient of development cost reduction for CGI may be discrepant for the two partners because of their difference of knowledge input level, learning ability, and collaboration level. Following Arsenyan [14], the development cost structure functions of the two firms are expressed as

$$c_F = c_F - d_F(q_F + \theta m_F), c_S = c_S - d_S(q_S + \theta m_S)$$

In conclusion, the function of green innovation revenue for two the firms are expressed as

$$\pi_F = \omega\left(a + b\bar{z}\frac{v+1}{2}\right) - (1-k)I\theta^2 - q_F - \left[c_F - d_F(q_F + \theta m_F)\right] \tag{8}$$

$$\pi_S = (1 - \omega)\left(a + b\overline{z}\frac{v+1}{2}\right) - kI\theta^2 - q_S - \left[c_S - d_S(q_S + \theta m_S)\right] \tag{9}$$

### 3.4. Model Solution by Using Nash Bargaining Equilibrium

Based on Nash bargaining game [54], collaborative parties bargain with each other in games until an agreement is achieved, and then partners act cooperatively for mutual benefit by following the agreement. Otherwise, parties act non-cooperatively. In our case of CGI, firstly the manufacturer calculates and proposes a level of innovation cooperation, based on its analysis of costs and benefits from the CGI. Then, the supplier calculates the innovation cost-sharing ratio. Finally, the two parties adjust the green profit distribution ratio by using Nash bargaining game, and the equilibrium solution should be achieved at maximizing the overall revenue of two parties. The game process is described as follows:

(1) Based on the revenue function, the manufacturer calculates the optimal CGI level $\theta^*$, which maximize the $\pi_F$ value in the formula (8).

$$
\begin{aligned}
\pi_F &= \omega\left\{a + b\tfrac{v+1}{2}\left[q_F + q_S + (m_F + m_S)\theta\right]\right\} - (1-k)I\theta^2 - q_F \\
&\quad - \left[c_F - d_F(q_F + \theta m_F)\right] \\
&= \omega\left[a + b\tfrac{v+1}{2}(q_F + q_S) + b\tfrac{v+1}{2}(m_F + m_S)\theta\right] - q_F \\
&\quad - \left[(c_F - d_F q_F) - d_F m_F \theta\right] - (1-k)I\theta^2
\end{aligned}
$$

Through substituting variables with $\overline{a} = a + b\tfrac{v+1}{2}(q_F + q_S)$, $\overline{b} = b\tfrac{v+1}{2}(m_F + m_S)$, $\overline{c_F} = c_F - d_F q_F$, and $\overline{d_F} = d_F m_F$, we get simplified equation as:

$$
\begin{aligned}
\pi_F &= \omega(\overline{a} + \overline{b}\theta) - (\overline{c_F} - \overline{d_F}\theta) - (1-k)I\theta^2 - q_F \\
&= -(1-k)I\theta^2 + (\omega\overline{b} + \overline{d_F})\theta + \omega\overline{a} - \overline{c_F} - q_F
\end{aligned}
$$

By solving first derivative function $\frac{\partial \pi_F}{\partial \theta} = (\omega\overline{b} + \overline{d_F}) - 2(1-k)I\theta = 0$, we get

$$\theta^* = \frac{\omega\overline{b} + \overline{d_F}}{2(1-k)I}$$

(2) Based on the CGI level $\theta^*$ proposed by the manufacturer, the supplier calculates the optimal cost-sharing rate $k^*$ through formula (9).

$$\pi_S = \left[(1-\omega)\,\overline{a} - \overline{c_S}\right] + \left[(1-\omega)\overline{b} + \overline{d_S}\right]\theta - q_S - kI\theta^2$$

Through variable substitutions with

$$\overline{e} = (1-\omega)\overline{a} - \overline{c_S}, \ \overline{f} = (1-\omega)\overline{b} + \overline{d_S}, \ \overline{g} = (\omega\overline{b} + \overline{d_F}),$$

we get

$$\pi_S = \overline{e} + \frac{\overline{f}\,\overline{g}}{2(1-k)I} - \frac{k\overline{g}^2}{4I(1-k)^2} - q_S$$

and let

$$\widetilde{k} = (1-k)$$

We get:

$$\pi_S = -\frac{(1-\widetilde{k})\overline{g}^2}{4I\widetilde{k}^2} + \frac{\overline{f}\,\overline{g}}{2I\widetilde{k}} + \overline{e} - q_S$$

Let

$$\widetilde{k} = (1-k)$$

We get:

$$\pi_S = -\frac{(1-\widetilde{k})\overline{g}^2}{4\widetilde{Ik}^2} + \frac{\overline{fg}}{2\widetilde{Ik}} + \overline{e} - q_S.$$

Let

$$\frac{\partial \pi_S}{\partial \widetilde{k}} = \frac{\overline{g}^2(2\widetilde{k} - \widetilde{k}^2)}{4\widetilde{Ik}^4} - \frac{\overline{fg}}{2\widetilde{Ik}^2} = 0$$

We get:

$$\widetilde{k} = 0$$

Or

$$\widetilde{k} = \frac{2\overline{g}^2}{2\overline{fg} + \overline{g}^2} \tag{10}$$

And then

$$k = (1 - \widetilde{k}) = \frac{2\overline{f} - \overline{g}}{2\overline{f} + \overline{g}}$$

Then, we get

$$\kappa^* = \frac{(2 - 3\omega)b(v + 1)(m_F + m_S) + 4d_S m_S - 2d_F m_F}{(2 - \omega)b(v + 1)(m_F + m_S) + 4d_S m_S + 2d_F m_F}$$

$$(1 - \kappa^*) = \frac{2\omega b(v + 1)(m_F + m_S) + 4d_F m_F}{(2 - \omega)b(v + 1)(m_F + m_S) + 4d_S m_S + 2d_F m_F}$$

(3)  Based on the Nash bargaining game, the two partners determine the profit distribution ratio $\omega^*$. As the collaboration aim is to maximize the anticipative total green revenue $R(\omega) = \pi_F + \pi_S$, we obtain the objective function in function (11). That is, for any $\omega$, $0 \leq \omega \leq 1$,

$$R(\omega) = \max(\pi_F + \pi_S) \tag{11}$$

Based on Nash bargaining game, the constraint condition is just need to max $\pi_F * \pi_S$ at proper profit distribution ratio. Here, by bringing $\kappa^*$ and $\theta^*$ into formulas (8) and (9), we can obtain the respective revenue function for the two firms, which are two complicated functions about revenue distribution ratio, and here we omit them. Finally, substituting these into Nash bargaining anticipative function $F(\omega) = \pi_F * \pi_S$, we can solve optimal $\omega$.

## 4. Simulation in Different Scenarios

In this section, we aim to analyze the optimal profit distribution ratio and total CGI revenue under different scenarios, and assess the influence of the above discussed factors on the total CGI revenue. Firstly, we conducted an investigation on automobile enterprises CGI in Henan province, China, and then set up the parameters for the discussed factors which lead to different scenarios. Meanwhile, we consider linguistic values such as high or low for the involved parameters. Finally, we test ten scenarios. These parameters include: (1) Trust levels between the two parties (from low to high), (2) knowledge complementarity (from low to high), (3) innovation efficiency, and (4) decreasing coefficient of development cost in green innovation. By taking these parameters as shown in Table 2, we simulate the expected function $F(\omega)$ in Matlab. The results are graphically presented from Figures 1–10, with *x*-axis representing profit distribution ratio and y-axis representing for the Nash total CGI revenue $F(\omega)$.

**Table 2.** Parameters for different scenarios in collaborative green innovation (CGI).

| Scenarios | $t_{FS}, t_{SF}$ | $q_F, q_S$ | $c_F, c_S$ | $d_F, d_F$ | $\beta$ | a, $\nu$ |
|---|---|---|---|---|---|---|
| Figure 1 | 0.2, 0.2 | 1100, 1000 | 3000, 3000 | 0.05, 0.05 | 0.8 | 0, 0.6 |
| Figure 2 | 0.4, 0.4 | 1100, 1000 | 3000, 3000 | 0.05, 0.05 | 0.2 | 0, 0.6 |
| Figure 3 | 0.6, 0.6 | 1000, 100 | 3000, 3000 | 0.05, 0.05 | 0.8 | 0, 0.6 |
| Figure 4 | 0.8, 0.8 / 0.8, 0.2 | 1000, 100 | 3000, 3000 | 0.05, 0.05 | 0.2 | 0, 0.6 |
| Figure 5 | 0.8, 0.8 | 1100, 1000 | 3000, 3000 | 2, 2 | 0.2 | 0, 0.6 |
| Figure 6 | 0.8, 0.8 | 1000, 200 | 3000, 3000 | 2, 2 | 0.4 / 0.6 | 0, 0.6 |
| Figure 7 | 0.8, 0.8 | 200, 200 | 300, 3000 | 2, 2 | 0.8 / 0.9 | 0, 0.6 |
| Figure 8 | 0.8, 0.8 | 1000, 1000 | 3000, 3000 | 0.05, 0.05 | 0.6 | 0, 0.2<br>0, 0.8<br>$a > b\overline{z}\nu$, 0.2<br>$a > b\overline{z}\nu$, 0.8<br>$a < b\overline{z}\nu$, 0.2<br>$a < b\overline{z}\nu$, 0.8 |
| Figure 9 | 0.8, 0.8 | (50, 100, 200, 400, 600, 800, 900, 1000), 1000 | 3000, 3000 | 0.05, 0.05 | 0.6 | 0, 0.6 |
| Figure 10 | 0.8, 0.8 | (50, 100, 200, 400, 600, 800, 900, 1000), 1000 | 3000, 3000 | 2, 0.05 | 0.6 | 0, 0.6 |

## 4.1. The effect of trust on total CGI revenue

In Scenarios 1–4, we analyze effect of trust on total CGI revenue and the optimal profit distribution strategy. Scenario 1 is a case that knowledge input is high for both parties, but F is a bit higher than S, cost structure for green innovation is nearly similar, and mutual knowledge complementarity is high. As shown in Figure 1, it demonstrates that total CGI revenue is high when trust between the two parties is high, and the highest revenue is achieved when profit distribution slightly favors ($\omega = 0.6$) the manufacturer. Scenario 2 (Figure 2) represents the situation of a high knowledge input, but F is a bit higher than S, and a similar cost structure for the two parties, but a low knowledge complementarity. Similar to the case of Scenario 1, total CGI revenue is high when trust is high, and the optimal revenue is achieved when profit distribution slightly favors ($\omega = 0.6$) the manufacturer. But distinctly different from the case in Scenario 1, the total CGI revenue in Scenario 2 is significantly lower. Scenario 3 presents the situation in which knowledge input level is high only for the manufacturer, but low for the supplier. As shown in Figure 3, although total CGI revenue is significantly lower, trust still has a positive effect on total revenue. Under this scenario, the optimal strategy is that the manufacturer contributes more to the collaborative innovation and also retains more profit generated from the collaborative innovation. Scenario 4 assumes the situation in which knowledge input of the manufacturer significantly high, cost structure of the two firms is similar, and knowledge complementarity is low. In this scenario, trust still has a positive but slight effect on total CGI revenue (because lines are denser), and total CGI revenue is maximized when the manufacturer retains more of the profit ($\omega = 0.8$).

In summary, when knowledge input is high for both parties, trust positively affects total CGI revenue no matter if knowledge complementarity level is high or low; when knowledge input is high only for one party (the manufacturer), influence of trust on total CGI revenue is rather marginal, although still positive, and the total CGI revenue is maximized when the party with a higher input also retains more collaborative profit.

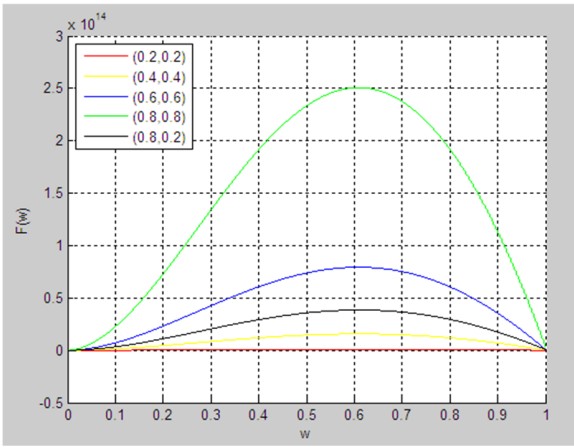

**Figure 1.** Effects of trust on total CGI revenue.

Scenario 1: Knowledge input is high for both parties but F is a bit higher than S, cost structure is similar, mutual knowledge complementarity is high. Mutual trust level is varying.

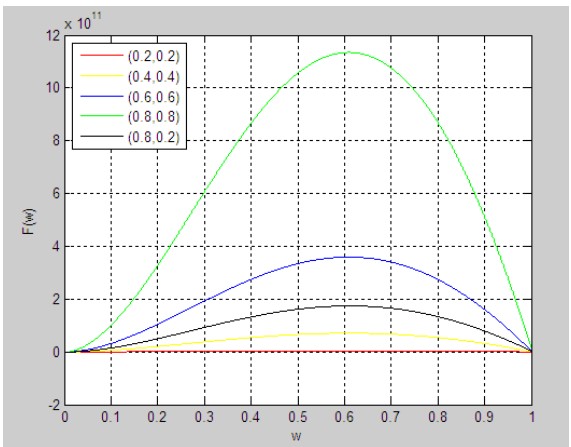

**Figure 2.** Effects of trust on total CGI revenue.

Scenario 2: Knowledge input is high for both parties, but F is a bit higher than S, cost structure is similar, mutual knowledge complementarity is low. Mutual trust level is varying.

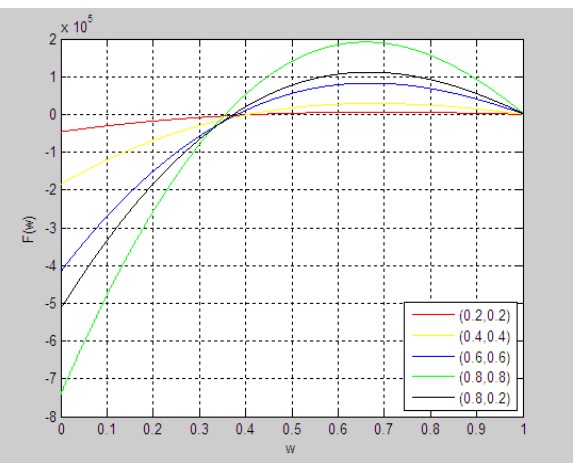

**Figure 3.** Effects of trust on total CGI revenue.

Scenario 3: Knowledge input is high for F, but low for S, cost structure is similar, mutual knowledge complementarity is high. Mutual trust level is varying.

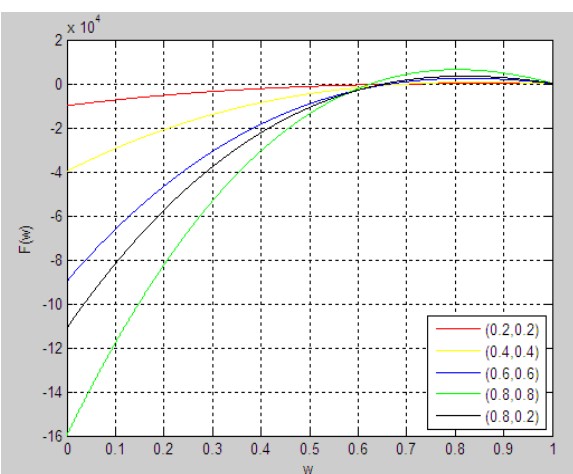

**Figure 4.** Effects of trust on total CGI revenue.

Scenario 4: Knowledge input is high for F, but low for S, cost structure is similar, mutual knowledge complementarity is low. Mutual trust level is varying.

*4.2. The Effect of Knowledge Complementarity on Total CGI Revenue*

Scenarios 5–7 assess the effect of knowledge complementarity on total CGI revenue. Scenario 5 represents the situation in which trust is high, knowledge input is high, but F a bit higher than S, initial development cost is equal, and rate of cost reduction is high for both parties. In this scenario, we can observe that when knowledge complementarity between the two firms is high, total CGI revenue is also high, which is maximized when profit ratio slightly favors ($\omega = 0.6$) the manufacturer. Scenario 6 shows that total CGI revenue is significantly reduced when the knowledge input from the manufacturer is much higher than the supplier, but total CGI revenue rises with increase of the knowledge complementarity. Moreover, when knowledge complementarity is at a low level ($\beta = 0.2$), total CGI revenue is very low, and the optimal strategy is when profit ratio is nearly equally distributed. The reason is that the manufacturer could not learn much from the supplier because of the low knowledge input from the supplier, while the supplier can learn much more from the manufacturer due to the high knowledge input from the manufacturer. It is the effect of learning that explains the fair distribution at different knowledge investment. With the complementarity increasing, learning level of the manufacturer would catch up with supplier, which indicates the manufacturer's contribution to total knowledge stock increases, so at the high knowledge complementarity ($\beta = 0.9$), total CGI revenue is maximized when the manufacturer has a significantly higher percentage of profit. Scenario 7 assesses the effect of knowledge complementarity on total CGI revenue under the conditions that knowledge input is low for both firms, initial development costs for manufacturer is low, but for supplier is high, the rate of cost reduction is also high. The simulation results show that the higher knowledge complementarity, the higher total CGI revenue, but which in this scenario is much lower in comparison to the previous scenarios. As illustrated by Figure 7, the total CGI revenue increases with increase of knowledge complementarity, which shows co-learning has an important effect for innovation. When knowledge complementarity is low ($\beta = 0.2$), there is less for the two parties to learn from each other. As the manufacturer has a lower level of initial development cost construct, the optimal strategy is that manufacturer retains a bit more percentage of the profit. When knowledge complementarity increases, (e.g., at 0.9 in Figure 7), the manufacturer has good cost structure in taking on more tasks, and then retains most of the profit ($\omega = 0.7$).

In summary, our simulation results demonstrate a positive effect of knowledge complementarity for total CGI revenue, especially in terms of when initial development cost is high.

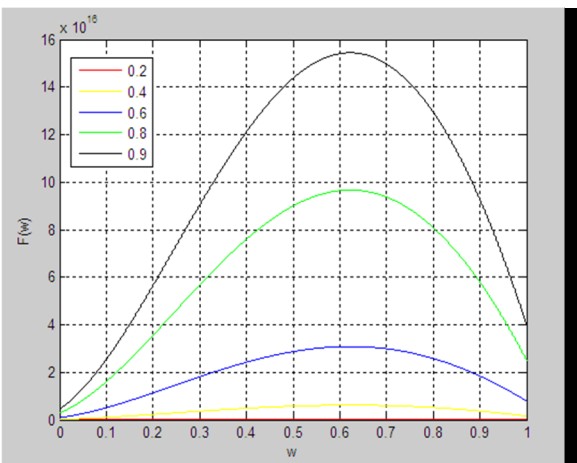

**Figure 5.** Effects of β on total CGI revenue.

Scenario 5: Mutual trust level is high, knowledge input is high, but F a bit higher than S, initial development cost is equal, and cost reduction rate is high for both parties. Knowledge complementarity is varying.

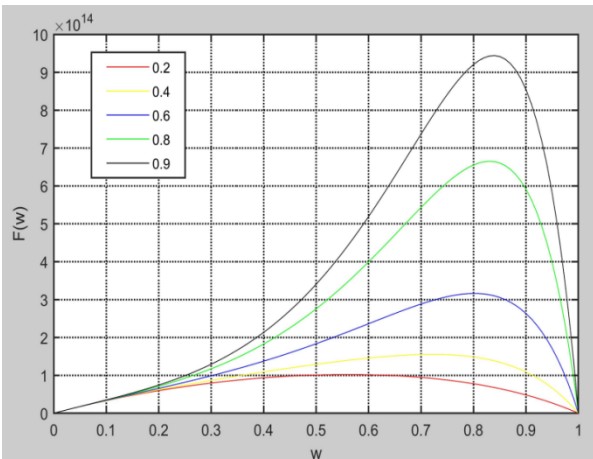

**Figure 6.** Effects of β on total CGI revenue.

Scenario 6: Mutual trust level is high, knowledge input is high for F, but low for S, initial development cost is equal, and cost reduction rate is high for both parties. Knowledge complementarity is varying.

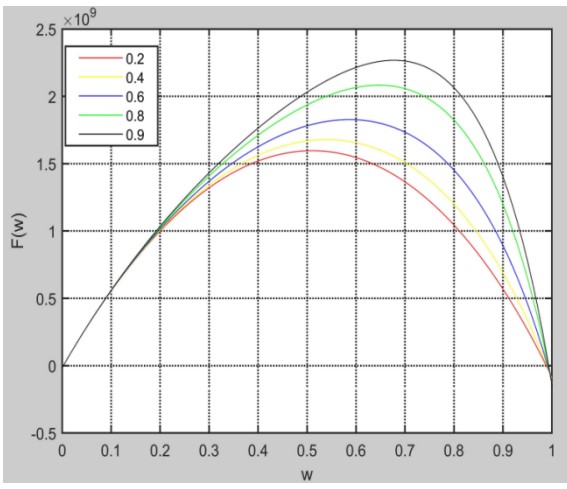

**Figure 7.** Effects of β on total CGI revenue.

Scenario 7: Mutual trust level is high, knowledge input is low for both firms. Initial development cost for manufacturer is low, but for supplier is high, the rate of cost reduction is also high. Knowledge complementarity is varying.

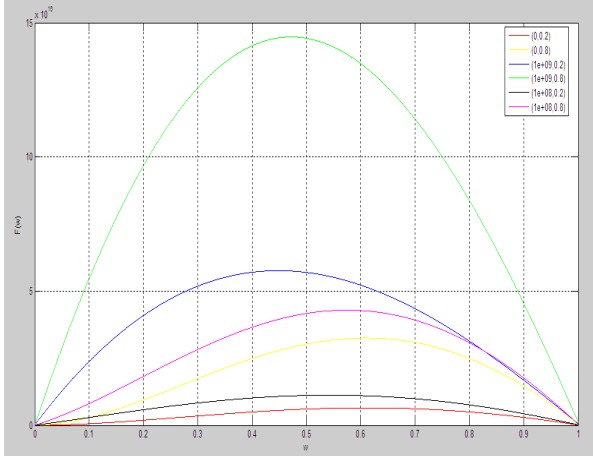

**Figure 8.** Effects of a, ν on total CGI revenue.

Scenario 8: Trust level is high, knowledge input and initial development cost are high, and cost reduction rate is low for both parties, knowledge complementarity is at moderate level.

### 4.3. The Effect of Product Type and Innovation Efficiency on Total CGI Revenue

To assess the effect of initial value, the added green value, and the innovation efficiency on total CGI revenue, we set up the parameters as: High level of mutual trust, high level of knowledge input and initial development cost, low rate of cost reduction, and moderate level of knowledge complementarity for the both parties. The six curves in Figure 8 illustrate the total CGI revenue under six different combinations of product type and innovation efficiency. (1) New product with low efficiency, (2) new product with high efficiency, (3) old product with minor innovation and low efficiency, (4) old product with minor innovation and high efficiency, (5) old product with major innovation and low efficiency, (6) old product with major innovation and high efficiency. As demonstrated in Figure 8, when the cost structures and knowledge stocks for both parties are equal, for old product with minor green innovation, whether high efficiency or not, the optimal profit ratio is equally distributed ($\omega = 0.5$) between the two partners. For the new product and old product with major innovation at

high efficiency, total CGI revenue reaches the maximum level when the manufacturer retains a higher share of the profit ($\omega = 0.6$). For low efficiency of new product and old product with major innovation, as collaborative green innovation can only generate a very low level of revenue, we can deduce that the risk of CGI is large, so the optimal strategy is a fair distribution ($\omega = 0.5$).

### 4.4. The Effect of Green Knowledge Input on Total CGI Revenue

To assess effect of green knowledge input by the supplier on total CGI revenue, we set up the parameters in Scenario 9 as high trust level between partners, high initial development cost and low cost reduction rate for both partners, moderate level of knowledge complementarity, and high level of knowledge input for the supplier. In this scenario, co-learning cannot reduce the development cost. The results demonstrate that the manufacturer increases its knowledge input, the total CGI revenue increases, and reaches the highest point when profit is equally distributed in every case. While input from the manufacturer equals to that of the supplier, total CGI revenue is maximized. While based on Scenario 10, if manufacturer has a high cost reduction rate, it represents that the manufacturer would retain a higher percentage of the profit with the increasing of its knowledge input, in that it has better cost structure.

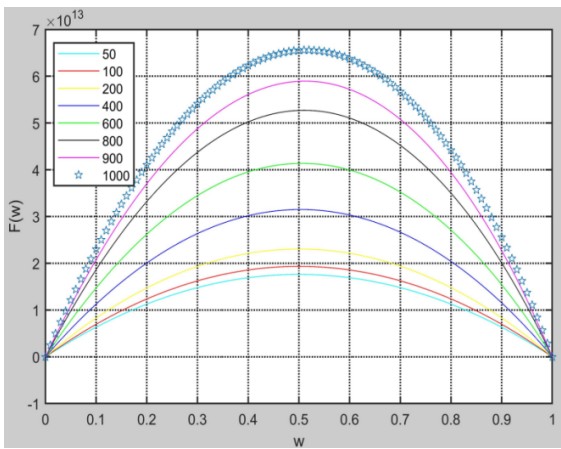

**Figure 9.** Effects of $q_F$ on total CGI revenue.

Scenario 9: Mutual trust level and initial development cost is high, cost reduction rate is low for both parties. Knowledge complementarity is at moderate level. High level of knowledge input for the supplier, but for manufacturer is varying.

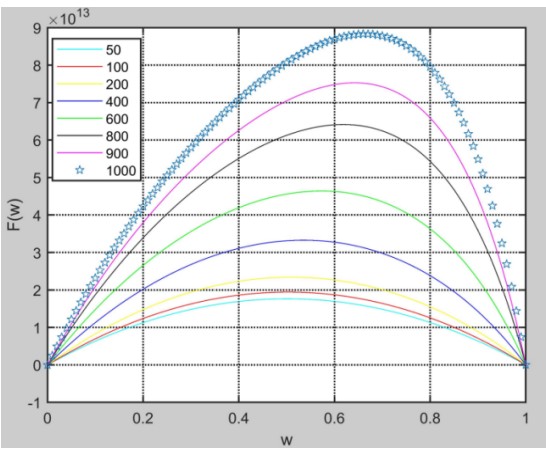

**Figure 10.** Effects of $q_F$ on total CGI revenue.

Scenario 10: Mutual trust level and initial development cost is high for both parties. Cost reduction rate is low, knowledge complementarity is at moderate level. High level of knowledge input for the supplier, but for manufacturer is varying.

## 5. Discussion and Conclusions

Applying game theory principles, our study investigated the decision-making process of the CGI between a focal manufacturer and its key supplier, and presented a mathematical model that captures the relation between total revenue of collaborative green innovation and profit distribution ratios of the two partners. Incorporating the five key factors of trust level, initial knowledge input, development cost, knowledge complementarity, and collaborative innovation efficiency into the mathematical model, our model provides an explanation of the mechanisms through which the key factors affect decision-making of the collaborative green innovation, expressed as parameters, and puts forward an analytic tool for green innovation practitioners to understand how the important collaboration dynamics of the individual influencing factors are when making decision to form an collaboration in green innovation. While existing game models of collaboration are mainly built on the profit sharing basis between partners and thus have paid more attention to the monetary gains of the collaboration partners [10,12,17], our game model puts more emphasis on providing a mathematical tool for green innovation practitioners to understand how important the collaboration dynamics are when negotiating and progressing a CGI project and on generating collaborating outcomes of a partnership regarding mutual learning and innovation levels generated from CGI with regard to the assessed parameters such as trust, knowledge complementarity, product type, and innovation efficacy. Our simulation results demonstrate that while it is important for both collaboration parties of the manufacturer and supplier to receive profit, a focus only on the profit sharing is not enough, as the central purpose of CGI is to add green value both environmentally and financially to the product through collaborative innovation. Thus, it is at least equally essential that the two parties pool their complementary knowledge together in order to extract new knowledge results through adding value for the green innovation. More specifically, our analytical developments and supporting numerical simulation provide a number of insights regarding the key dimensions in affecting inter-firm collaboration of green innovation, and they include:

First, applying game theory approach to assess the bargaining conditions for decision-making to form partnership of CGI, our modelling explored the role of the profit-sharing ratio in coordinating collaboration level, and stabilizing the partnership, and influencing collaboration efficacy and total value creation of the collaboration. Our simulation results suggest that an equal ratio for profit sharing between the two partners is not optimal with regarding to mutual learning and value generation from the CGI under certain situations, even when the levels of green knowledge input, initial cost structure, and reduction rate of cost are similar or the same for the manufacturer and supplier (Scenario 8).

Second, as the key factor for a successful partnership in supply chain relationship, trust plays a crucial role in determining involvement of the supplier in CGI and in mutual decision-making regarding collaboration level. Without trust, collaboration between the partners becomes not preferable. Implications from this result is that the manufacturer should make full use its relational capital and social network to initiate and develop long term trust-based partnership for the mutual benefits of the involved two partners, as well as society as a whole.

Third, knowledge complementarity provides potential opportunity for mutual learning between the two collaboration partners, and absorptive capacity enables knowledge-sharing and knowledge creation by serving as a bridge between the knowledge inputs from the partners. Our modelling simulations confirmed the role played by mutual learning in forming a successful partnership for CGI. Through enabling effective mutual learning, knowledge complementarity and knowledge input facilitate knowledge creation in relating to green innovation and development of green products, leading to a higher level of collaboration revenue. An implication from this research insight is that as the leading firm in the collaborative innovation partnership, the focal manufacturer needs to select

a supplier with which the knowledge stock of manufacturer can complement, so that the efficiency of CGI can be increased.

Fourth, whether the green innovation is for new product development or upgrading of existing product, whether the innovation efficacy is high or low, on what ratio the anticipative collaboration is distributed between the collaborative partners also have significant influence on the level of collaboration revenue. Our modelling simulation confirmed that the collaborative projects for combinations of old product with minor innovation with either low or high efficacy tend to generate a high level of collaboration revenue, and the revenue would reach maximum when the revenue is equally distributed between partners. For the combination of new product or old product with major innovation at high efficacy, the total CGI revenue reaches maximum level when the manufacturer retains a higher level of profit. For low efficiency of new product and old product with major innovation, the optimal strategy is a fair share.

In the end, we would like to discuss a few limitations of our model, leading to potential opportunities for future research. First, our simulation analysis has not provided clear-cut ranges for the parameters assessed. The levels given for the variables of trust, knowledge complementarity, new product type, and innovation efficacy, are defined in a linguistic echelon. Future research could test the effectiveness of our game model with an empirical data set collected from the industry. Second, we assume that the manufacturer acts as a leading partner that has a higher level of bargaining power, and thus potentially a higher proportion of the generated CGI profit. In reality, CGI could be initiated by either the manufacturer or the supplier. The supplier could negotiate with the manufacturer based on a higher bargaining power. Future research could address this issue. Third, our model is limited to the supply chain relationship between a single manufacturer and a single supplier. In the future work, we will consider CGI in the context of a manufacturer and multiple suppliers.

**Author Contributions:** Conceptualization, Q.L. and Y.K. Data curation, L.T. and B.C. Funding acquisition, Q.L. Investigation, L.T. Methodology, Q.L. and Y.K. Project administration, Q.L. Resources, Y.K. Supervision, Y.K. Visualization, B.C. Writing—original draft preparation, Q.L. and Y.K. Writing—review & editing, Q.L. and L.T. All authors have read and agreed to the published version of the manuscript.

**Funding:** This research was financially supported by the Humanities and Social Science Foundation, Ministry of Education in China (grant No. 16YJC630053), National Social Science Foundation of China (grant No.19CGL004), Shaanxi Natural Science Foundation (grant No. 2019JQ-692), China Postdoctoral Science Foundation (grant No. 2015M572531), The Central Universities Fundamental Research Funds for Chang'an University (grant No. 310823170654, 300102230652, 300102239605), Shaanxi Education Department Project (grant No. 19JZ011).

**Conflicts of Interest:** The authors declare no conflict of interest.

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
