# Peer review of "Modeling Formation and Operation of Collaborative Green Innovation between Manufacturer and Supplier: A Game Theory Approach"

_sustainability, doi:10.3390/su12062209_

Round 1

Reviewer 1 Report

Very well written paper. I do not feel qualified to evaluate the math and simulations, but the authors have a very good writing style and well articulated presentation. 

Author Response

Response: Thank you for the positive comments about writing and presentation of our manuscript, which are quite encouraging for us to address the important topic of our study. We believe this is an important topic, which has not been adequately addressed and thus needs have more research attention.

Reviewer 2 Report

Review Report #   sustainability-728646

This study explores collaborative green product manufacturing issues. The topic is important; however, the authors need to revise the manuscript based on the following comments.

  1. The literature review needs to improve. The authors may find the following articles in this direction:

Comparative analysis of government incentives and game structures on single and two-period green supply chain, Journal of Cleaner Production 235, 1371-1398

The impact of strategic inventory and procurement strategies on green product design in a two-period supply chain, International Journal of Production Research 57 (7), 1915-1948

Is It a Strategic Move to Subsidized Consumers Instead of the Manufacturer?IEEE Access 7, 169807-169824

2.In Section 3.2, explain how the authors measure trust lave. Please provide practical examples in this direction.

3. Rewrote equations (10) and (11). Without the centralized decision (integrated supply chain), it is difficult to understand how much performance is improved. Therefore, it is suggested to derive optimal decision under the centralized scenario and compared the results

4. Please provide the detail caption for Fig. 1-4 and other also, only scenario does not make any insights. Moreover, they look like similar in nature

Author Response

This study explores collaborative green product manufacturing issues. The topic is important; however, the authors need to revise the manuscript based on the following comments.

Response: Thank you for the valuable comments and suggestions which have been a great help to us in improving our manuscript. For convenience, we reproduce each of your comments below followed in turn by our responses. Explanations and revisions are set out in italic.

  1. The literature review needs to improve. The authors may find the following articles in this direction:

Comparative analysis of government incentives and game structures on single and two-period green supply chain, Journal of Cleaner Production 235, 1371-1398

The impact of strategic inventory and procurement strategies on green product design in a two-period supply chain, International Journal of Production Research 57 (7), 1915-1948

Is It a Strategic Move to Subsidized Consumers Instead of the Manufacturer? IEEE Access 7, 169807-169824

Response: In responding to the review comments, we have also improved the structure of literature review/theoretical background in introduction section. We would like to provide an explanation here to provide a rationale and the structure for our literature review.

Our literature review and introduction sections aim to achieve the following four objectives as:

  • By positioning our research topic in the two literature areas of supply chain research and environmental sustainability, we have reviewed relevant prior studies in the cross-discipline manner. Based on the review of literature, we have adopted the transaction cost theory as the theoretical approach to address the identified research gap.

  • Based on a review of the game theory and modelling methods used in behavior research of supply chain green innovation activity, we have introduced the game theory and its models as the methodological approach to develop our game models and to run numerical simulation analysis of formation and operation of collaborative green innovation between the manufacturer and supplier.

  • Based on the review of social exchange theory as the theoretical lens, we have drawn relevant influencing factors for the formation and success of collaborative green innovation, and

We are also grateful to your great help in providing the useful references. In our revisions, all the three recommended references have been cited in the manuscript. The full references are provided as below:

Nielsen, I. E., Majumder, S., Sana, S. S., & Saha, S. (2019). Comparative analysis of government incentives and game structures on single and two-period green supply chain. Journal of Cleaner Production235, 1371-1398.

Dey, K., Roy, S., & Saha, S. (2019). The impact of strategic inventory and procurement strategies on green product design in a two-period supply chain. International Journal of Production Research57(7), 1915-1948.

Saha, S., Majumder, S., & Nielsen, I. E. (2019). Is It a Strategic Move to Subsidized Consumers Instead of the Manufacturer? IEEE Access7, 169807-169824.

  1. In Section 3.2, explain how the authors measure trust lave. Please provide practical examples in this direction.

Response: Based on the trust-commitment theory, prior studies have provided a measure for the concept of trust (Cheng, Yeh & Tu, 2008; Panteli et al, 2005). We have adopted this measure of trust in our study. This measure is provided as below:

The trust-commitment theory suggests that a successful partnership requires commitment among the partners and that trust is a critical elements to sustain such commitment. Thus, trust can be measured through by two dimensions of transaction cost variables (including asset specificity and behavioral uncertainty) and social exchange variables (including perceived satisfaction, reputation, and conflict).  

The references from which we draw out measure of trust are provided as below:

Panteli, N. Sockalingam, S. Trust and conflict within virtual inter-organizational alliances: a framework for facilitating knowledge sharing, Decision Support Systems, 2005, 4, 599–617.

Cheng, J.H., Yeh, C.H. Tu, C.W. Trust and knowledge sharing in green supply chains. Supply Chain Management: An International Journal, 2008, 4, 283-295.

  1. Rewrote equations (10) and (11). Without the centralized decision (integrated supply chain), it is difficult to understand how much performance is improved. Therefore, it is suggested to derive optimal decision under the centralized scenario and compared the results.

Response: Our model is constructed by considering the partnership of the collaboration green innovation as the basis. The objective function is to maximize the total revenue of collaboration green innovation for the two involved parties, which is a joint decision question by stepwise strategy based on the Nash Game. In order to develop a clearer equation expression, we integrated a new total green revenue function by merging equations (10) and (11).

  1. Please provide the detail caption for Fig. 1-4 and other also, only scenario does not make any insights. Moreover, they look like similar in nature.

Response: Yes, we agree the comments here that the Figures look similar with no a clear description. Following the review comments, we have provided captions with more detailed description for Figures 1-10. With the information provided on the captions, we hope that it would be easier to understand the simulation scenarios presented by these figures.

Constrained by the length of the manuscript, it is difficult for us to provide more details regarding our modelling computations and implementation. On the other hand, we will be very happy to provide these details to any interested reader, if being requested.

Reviewer 3 Report

Zagadnienia poruszone w artykule sÄ… ważne i nieczÄ™sto spotykane w literaturze. Przedstawiona metoda jest jednak bardzo zÅ‚ożona i dostÄ™pna tylko dla stosunkowo niewielkiej grupy odbiorców. Dlatego konieczne jest uproszczenie sekcji metod i opracowanie dokÅ‚adniejszych modeli używanych do opisu. 

The issues raised in the article are important and not often found in the literature. However, the presented method is very complex and available only to a relatively small group of recipients. Therefore, it is necessary to simplify the method sections and develop more accurate models used for the description.

Author Response

Response: We are encouraged by the positive comments regarding our research topic. Following the suggestion, we have tried to develop clearer and simpler statements and expressions regarding the method adopted for our game modelling analysis. For example, we integrated equations 10 and 11 with the aim to have a clearer description of modelling method. We hope that these simplified modelling expressions are able to facilitate a better understanding by our readers.

Round 2

Reviewer 2 Report

The authors made an effort to improve the paper, I recommended acceptance.

Reviewer 3 Report

I have no comments